

# Development of the adjoint of the GEOS-Chem unified tropospheric-stratospheric chemistry extension (UCX) in GEOS-Chem Adjoint v36

Irene C. Dedoussi[1,2], Daven K. Henze[3], Sebastian D. Eastham[2], Raymond L. Speth[2], Steven R.H. Barrett[2]

[1]Section Aircraft Noise and Climate Effects, Faculty of Aerospace Engineering, Delft University of Technology, Kluyverweg 1, 2629 HS, Delft, The Netherlands
[2]Laboratory for Aviation and the Environment, Department of Aeronautics and Astronautics, Massachusetts Institute of Technology, 77 Massachusetts Avenue, Cambridge, MA 02139, United States
[3]Department of Mechanical Engineering, University of Colorado Boulder, 1111 Engineering Drive, Boulder, CO 80309, United States

*Correspondence to*: Irene C. Dedoussi (i.c.dedoussi@tudelft.nl)

**Abstract.** Atmospheric sensitivities (gradients), quantifying the atmospheric response to emissions or other perturbations, can provide meaningful insights on the underlying atmospheric chemistry or transport processes. Atmospheric adjoint modelling enables the calculation of receptor-oriented sensitivities of model outputs of interest to input parameters (e.g. emissions), overcoming the numerical cost of conventional (forward) modelling. The adjoint of the GEOS-Chem atmospheric chemistry-transport model is a widely used such model, but prior to v36 has lacked extensive stratospheric capabilities. Here, we present the development and evaluation of the discrete adjoint of the global chemistry transport model (CTM) GEOS-Chem unified chemistry extension (UCX) for stratospheric applications, which extends the existing capabilities of the GEOS-Chem adjoint to enable the calculation of sensitivities that include stratospheric chemistry and interactions. This development adds 37 new tracers, 273 kinetic and photolysis reactions, an updated photolysis scheme, treatment of stratospheric aerosols, and all other features described in the original UCX paper. With this development the GEOS-Chem adjoint model is able to capture the spatial, temporal and speciated variability in stratospheric ozone depletion processes, among other processes. We demonstrate its use by calculating two-week sensitivities of stratospheric ozone to precursor species and show that the adjoint captures the Antarctic ozone depletion potential of active halogen species, including the chlorine activation and deactivation process. The spatial variations in the sensitivity of stratospheric ozone to $NO_x$ emissions are also described. This development expands the scope of research questions that can be addressed, by allowing stratospheric interactions and feedbacks to be considered in the tropospheric sensitivity and inversion applications.



## 1 Introduction

Chemistry-transport models (CTMs) that simulate the chemistry, transport and deposition processes in the atmosphere provide a tool to investigate the atmospheric impact of current emissions, as well as emissions scenarios resulting from technological or policy decisions  Global CTMs that simulate both the troposphere and the stratosphere (and capture interactions between the two) can be employed to calculate stratospheric ozone, which plays a critical role in absorbing incoming solar ultraviolet (UV) light that could otherwise be harmful to human health, animals, plants, biogeochemistry, air quality, and materials (WMO/UNEP, 2014). Examples of such are the GEOS-Chem UCX (Eastham et al., 2014), MOZART-3 (Kinnison et al., 2007), TM5 (Huijnen et al., 2010), GMI (Considine et al., 2000; Rotman et al., 2001), OSLO-CTM3 (Søvde et al., 2012), and EMAC (Sausen et al., 2010).  GEOS-Chem is a three-dimensional global CTM originally developed by (Bey et al., 2001), and updated (http://www.geos-chem.org) with the unified chemistry extension (UCX) (Eastham et al., 2014). It has been used to quantify a variety of ozone-related mechanisms and impacts including those of aviation-related ozone (Eastham and Barrett, 2016; Quadros et al., 2020), stratospheric ozone intrusions (Greenslade et al., 2017), and accelerated stratospheric ozone loss (Eastham et al., 2018), as well as processes of other atmospheric constituents such as halogens (Wang et al., 2019; Zhu et al., 2019; Sherwen et al., 2016).

The atmospheric parameters that affect and control the behaviour of the ozone layer can be assessed through sensitivity analyses. As explained by Hakami et al. (2007) and Clappier et al. (2017), sensitivity analyses can be performed in a forward or backward (adjoint) manner. In the forward method, a perturbation is introduced in a parameter of interest (source), and sensitivities are propagated from the perturbed source into the various receptors/outputs. The methods in this category (one of which is finite difference, also known as "brute force") are efficient in simultaneously providing information about all receptors with respect to the perturbed parameter. This method, however, is constrained by numerical noise (e.g. cancellation errors) (Hakami et al., 2007). When assessing the impacts of various sources, this approach can also result in significant computational overhead. In the backward (or adjoint) sensitivity analysis, a perturbation in the receptor is propagated backwards in time and space through an auxiliary set of equations, thus linking the effect on a scalar model output (receptor) originating from multiple model parameters (sources). As a result, the adjoint sensitivity analysis provides simultaneous sensitivity information about a specific outcome with respect to all sources and parameters. For example, an adjoint evaluation could provide, in a single simulation, the effect of perturbations of any ozone precurcor species at any location in the computational domain on the total adjoint total global stratospheric ozone mass. Adjoints can be employed to calculate sensitivities of metrics of interest with respect to a number of parameters at machine precision in accordance with model chemistry and physics that would have been currently impracticable to calculate otherwise (e.g. Henze et al. (2008), Nawaz et al. (2023), Dedoussi et al. (2020)). Adjoint sensitivities



can also be used with gradient-based optimization algorithms (e.g. 4D-Var) to optimize model parameters and inputs (Kopacz et al., 2010; Qu et al., 2020).

The adjoint of the GEOS-Chem CTM was developed by Henze et al. (2007) with several updates since (Capps et al., 2012; Gu et al., 2023; Tang et al., 2023; Wang et al., 2012 and others). Albeit having extensive tropospheric chemistry capabilities, prior to v36 stratospheric processes are calculated based on archived data or simplified parameterizations (similar to the GEOS-Chem model capabilities before the introduction of the UCX). Stratospheric ozone is calculated using the linearized ozone parameterization (Linoz) scheme (Singh et al., 2009). The evolution of most other species in the stratosphere is calculated from production and loss rates archived from NASA's Global Modeling Initiative (GMI) code (Murray et al., 2012; Rotman et al., 2001). Finally, given that the v35 of the adjoint of GEOS-Chem is troposphere-focused, tracers necessary for detailed stratospheric calculations (e.g. chlorofluorocarbons (CFCs), water and methane) and processes (incl. stratospheric aerosols, polar stratospheric clouds, emissions from long-lived species) are also not present in pre-UCX versions.

Stratospheric processes and impacts however represent a crucial component of atmospheric chemistry, necessitating models to be able to quantify them and better understand them. Ozone has direct effects on human health, with exposure to UV light leading to an increased likelihood of eye damage and/or skin cancer (Slaper et al., 1996). After the discovery of the ozone depleting effect of industrially produced CFCs and halons over the Antarctic ('ozone hole') as well as significant losses in other latitudes, the nations of the world agreed to protect the ozone layer under the 1987 Montréal Protocol and its amendments (Farman et al., 1985; Molina and Rowland, 1974; Solomon, 1999; McElroy et al., 1986; Solomon et al., 1986). In addition to CFCs, high altitude emissions (including volcanic emissions), climate change, and sunlight affect stratospheric ozone and need to be considered in modelling of stratospheric chemistry. While the Antarctic ozone hole has shown signs of recovery, the ozone layer remains an environmental topic of discussion (Ball et al., 2018; Solomon et al., 2016; Kuttippurath and Nair, 2017), as technological changes, industrial chemicals, and climate change could have a direct effect on stratospheric ozone depletion. There has been recent interest in supersonic commercial aircraft that cruise at ~50,000 ft, emitting nitrogen oxide ($NO_x$), which is also known to contribute to ozone depletion (Johnston, 1971; Crutzen, 1970; Cunnold et al., 1977; Eastham et al., 2022). Further, high altitude aviation emissions are known to change the ozone vertical distribution in the atmosphere (Eastham and Barrett, 2016; Köhler et al., 2008; Emmons et al., 2012; Brasseur et al., 1998; Maruhashi et al., 2022). Aviation emissions and corresponding impacts are expected to increase given that aviation is the transportation sector with the highest growth rate, with no direct replacement alternative (Schäfer et al., 2009). In addition, an increasing number of rocket launches and/or higher altitude launch payloads could also lead to higher emissions at stratospheric levels (Ross et al., 2009; Ryan et al., 2022). Industrial chemicals, in the form of short-lived chlorine species not controlled by the Montréal Protocol, have also been highlighted in terms of their ozone depletion potential (Hossaini et al., 2015, 2017). At the same time, CFC-11 and CFC-12 emissions, controlled by the Montréal Protocol, are found to be unexpectedly increasing (Rigby et al., 2019; Lickley et al., 2020; Montzka et al., 2018). Finally, the projected cooling of the stratosphere under increased greenhouse gas emission



scenarios could affect ozone depletion potentials, and thereby the recovery of the ozone hole (Weatherhead and Andersen, 2006).

In this paper we describe an alternative way of quantifying the effects of perturbations in ozone depleting precursors, through
the development of the adjoint of GEOS-Chem UCX, which is the first adjoint model of a unified tropospheric stratospheric detailed chemistry-transport model. The paper is structured as follows: Section 2 describes the UCX model and its adjoint development and Section 3 provides the model evaluation. Section 4 presents an application of the newly developed capabilities, computing sensitivities of stratospheric ozone burden to ozone depleting precursor perturbations in the global domain. Section 5 summarises the paper and lists limitations of the model development and application.

## 2 Methods

An adjoint model consists of a base ('forward component') model, and its corresponding differentiated counterpart ('differentiated component'). The forward model on which the GEOS-Chem adjoint is based corresponds to GEOS-Chem v8-02-01 with several updates and bug fixes, whereas the GEOS-Chem UCX forward model is version v10-01. The development in this work entails first updating the forward component in the adjoint to match the GEOS-Chem UCX capabilities in v10-15   01, and subsequently developing the corresponding differentiated counterpart code. The capabilities of the forward UCX model that are incorporated into the forward component of the adjoint are outlined in Section 2.1 below. The development choices for the adjoint model are then described in Section 2.2.

### 2.1 The UCX model

The GEOS-Chem UCX, as described and validated by Eastham et al. (2014), introduced stratospheric capabilities to the global
GEOS-Chem CTM, without compromising the existing tropospheric capabilities. This section briefly details the implementation of the UCX into the forward component of the GEOS-Chem adjoint, in addition to auxiliary changes required for the UCX capabilities to function.

The vertical domain of the chemical solver is extended to the top of the stratosphere, corresponding to ~1 hPa or ~50 km, and
the vertical resolution in the stratosphere is increased to match that of the GEOS model. This corresponds to an additional ~30 vertical layers, resulting in full chemistry calculations being solved in 59 of the 72 total vertical grid layers. Thirty-seven new tracers, necessary for the stratospheric chemistry calculations, are added to the GEOS-Chem adjoint model, and are listed in Table S1 of the SI. These also include water vapor and methane, which are now chemically active species within the model. Surface mixing ratio boundary conditions are added for the newly introduced long-lived species. The existing Fast-J photolysis
scheme did not consider wavelengths shorter than 289 nm, since those wavelengths are attenuated above the tropopause (Bian and Prather, 2002). These wavelengths however are essential in stratospheric chemistry (Sander et al., 2000). The photolysis





scheme is thus updated to Fast-JX v7.0 which expands the spectrum analysed to 18 wavelength bins covering 177-850 nm, and extends the upper altitude limit to approximately 60 km (Bian and Prather, 2002; Fast-JX v7.0a). We add 217 kinetic reactions and 43 photolytic decomposition processes, to bring the existing chemical mechanism in the GEOS-Chem adjoint up-to-speed with the UCX forward model mechanism. The original UCX additions were designed to match the GMI

stratospheric chemistry mechanism (Rotman et al., 2001) and to update the rates to JPL10-06 (Sander et al., 2011). We use the Kinetic PreProcessor (KPP) software library, version 2.2.3, to automatically generate the chemical mechanism (Damian et al., 2002; Sandu et al., 2003; Daescu et al., 2003) and include it in the model. Polar stratospheric cloud-related reactions are also added. Other minor additions (e.g. mesospheric $H_2SO_4$ photolysis, mesospheric $NO_x$ and $N_2O$ loss rates, etc.) and bug fixes introduced since the release of UCX are also added. Finally, we make no changes to global transport, convection or mixing

processes. Each individual change introduced is evaluated against the forward model (where possible). The entire model evaluation is described in Section 3.1.

## 2.2 Model development

To calculate adjoint sensitivities (gradients), the differentiated model needs to be generated from the base (forward component) model. Adjoint code can be derived in two ways – continuous and discrete. In a continuous adjoint, the

model governing equations are differentiated and then discretized for numerical solution. In such a case, the adjoint equations maintain their physical interpretability, but the algorithmic treatment may be very different from the forward component. In a discrete adjoint, the already discretized forward component is differentiated directly. Sandu et al. (2005) provide a description of discrete and continuous adjoints of CTMs. In this work a discrete adjoint approach is selected for the UCX adjoint, as it allows us to maintain algorithmic consistency with the GEOS-Chem UCX and thereby enables

direct validation.

Following the discrete adjoint approach entails generating the adjoint model (differentiated component) of the discretized forward component code. This is done through a combination of manually (by hand) derived and automatically generated differentiated code, using the Tapenade automatic differentiation (AD) tool (Hascoet and Pascual, 2013; Giering and Kaminski, 1998). Tapenade provides analytical derivatives of the computer program functions in cases where there is

significant variable interdependence and length of code. The PUSH/POP functions of Tapenade are utilised to store and retrieve (in reverse) recomputed forward model variable values that are needed in the adjoint equations. The adjoint of the chemistry mechanism is directly generated through KPP (Damian et al., 2002; Sandu et al., 2003; Daescu et al., 2003). In a model evaluation context, adjoint sensitivities are typically compared to finite difference sensitivities due to their ease of calculation, although potential errors introduced due to round-off, nonlinear effects, and discontinuities must be considered (Capps et al.,

2012; Giles and Pierce, 2000). Each individual adjoint subroutine is independently evaluated against forward model





sensitivities. The entire adjoint model evaluation is described in Section 3.2.

## 3 Model evaluation

First the implemented UCX model in the GEOS-Chem adjoint forward component is evaluated against the stand-alone GEOS-Chem UCX model. This evaluation is described in Section 3.1. Sensitivities from the differentiated counterpart of the adjoint
model are then evaluated against finite difference-based sensitivities from the forward model component. This evaluation is described in Section 3.2.

### 3.1 Base (forward component) model evaluation

To evaluate the performance of the forward model extensions in the GEOS-Chem adjoint model, we perform a five-year long simulation (January 1st 2008 –January 1st 2013). Five years is approximately the mean age of air in the upper stratosphere, measured from stratospheric entry at the tropical tropopause, and thereby a sufficiently long time to test whether the
stratospheric cycle is represented accurately in the model (Butchart, 2014). We perform three such simulations: one for the stand-alone GEOS-Chem UCX model (validated in Eastham et al. (2014)), one for the forward component of the GEOS-Chem adjoint v35f before the introduction of the UCX capabilities, and one for the forward component of the GEOS-Chem adjoint with the newly-introduced UCX extensions. The global grid has a horizontal resolution of 4° × 5° latitude and longitude respectively, and 72 vertical hybrid sigma-eta pressure levels extending from the surface to 0.01 hPa. The model is driven by
GEOS5 assimilated meteorological data from the Global Modeling and Assimilation Office (GMAO) at the NASA Goddard Space Flight Center. Identical initial conditions are used for all simulations. These are obtained by running the stand-alone GEOS-Chem UCX model for a 3-year time period (prior to the 5-year simulation) to 'spin-up' the model. Long-lived species are initialized based on archived zonal mean mixing ratios from the 2D stratospheric model AER CTM (Weisenstein et al.,
1997).





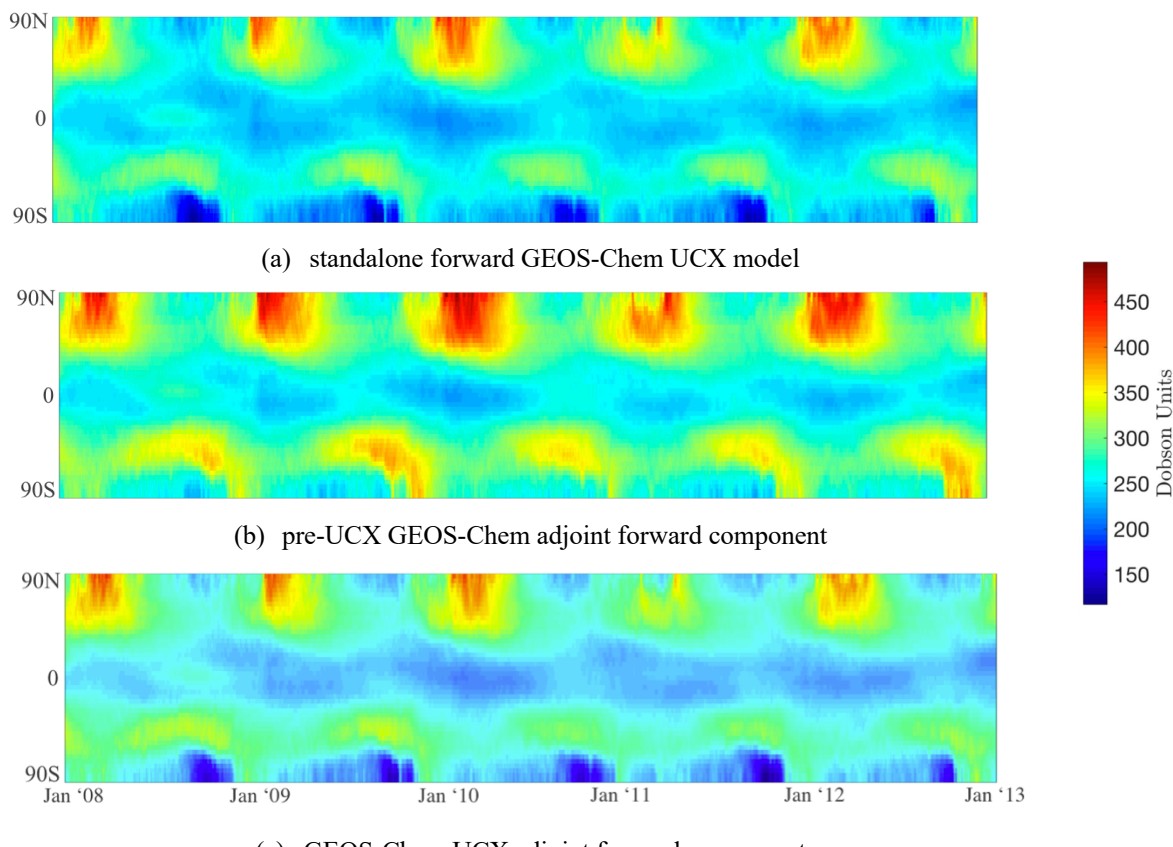

(a)   standalone forward GEOS-Chem UCX model

(b)   pre-UCX GEOS-Chem adjoint forward component

(c)   GEOS-Chem UCX adjoint forward component

**Figure 1: Zonal-mean column ozone for 2008-2012 from the standalone forward GEOS-Chem UCX model (a), the pre-UCX GEOS-Chem adjoint base (forward component) model (b) and the GEOS-Chem UCX adjoint the base (forward component) model (c). $O_3$ is shown for the forward UCX model and $O_x$ for the adjoint forward components**

Figure 1 shows the zonal-mean column ozone (in Dobson Units) for the 5-year long run for the forward UCX model, the forward component of the adjoint model ('pre-UCX' adjoint) and the forward component of the UCX adjoint model (developed as part of this work), shown in the top, middle and bottom subfigures respectively. Due to tracer differences between the models, $O_3$ is used for the forward UCX model and $O_x$ for the adjoint forward components. However, we note that the openly available v36 now uses $O_3$ as a tracer instead of $O_x$ (in addition to NO and $NO_2$ instead of $NO_x$). Similar to Eastham et al.

(2014), we use the ozone layer to demonstrate the improved stratospheric modelling, as it is a key feature of the stratosphere and sensitive to a variety of stratospheric processes (e.g. halogen cycles, aerosol formation, short-wavelength photochemistry, etc.). First, we are able to reliably reproduce the behaviour at mid-latitudes using online chemistry - this was achieved through relaxation to a known climatology in the pre-UCX adjoint forward component. This is also evident in Figure 2, which shows the mean ozone column as a function of latitude for 2010 for the three model versions. Second, the Antarctic ozone seasonal

cycle, a feature not captured in the 'pre-UCX' adjoint forward component, is now replicated (as in the standalone UCX model).

This is characterized by the formation of a deep "ozone hole" each September and the subsequent recovery by the end of the year (Solomon, 1999). This influences the rest of the southern hemisphere after the breakdown of the polar vortex each spring (Eastham et al., 2014). The mean zonal averaged absolute column ozone difference between the UCX standalone model and the UCX adjoint forward component is 2.7% for the 5-year run. Besides ozone, other key species have been compared, with the example of $NO_x$ included in the SI. These differences are to be expected as the UCX standalone model includes additional model updates and changes, beyond the UCX, that are not implemented in the adjoint forward component model.

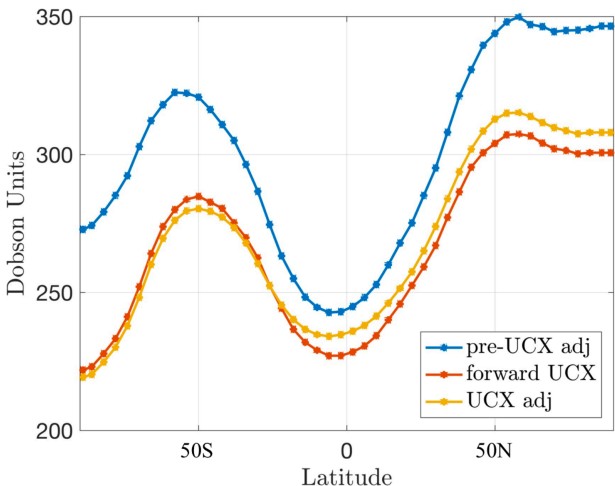

**Figure 2: Mean column ozone for 2010 for the standalone forward GEOS-Chem UCX model (red), the pre-UCX GEOS-Chem adjoint base (forward component) model (blue) and the GEOS-Chem UCX adjoint base (forward component) model (yellow).**

## 3.2 Adjoint model evaluation

The choice of a discrete adjoint allows the evaluation of the adjoint sensitivities directly against the forward component code (Giles and Pierce, 2000). While the performance of the adjoint UCX model has been evaluated in a component-wise manner for each individual module change or introduction (see SI Table S2 for the module list), here we evaluate the adjoint model as a whole for the generation of short-term sensitivities. To assess the accuracy of the adjoint modules constructed, adjoint sensitivities are compared with finite difference sensitivities from the forward component at the end of a single chemistry time-step (1 hour). To overcome the different nature of finite difference and adjoint sensitivities (source- and receptor oriented respectively), transport and convection processes are disabled so that each column of the 3D grid acts independently. This way $N$ evaluations are performed simultaneously, where $N$ is the number of grid cells in each layer of the horizontal grid ($N=46 \times 72$). In this column model test, the sensitivity of odd oxygen ($O_x$) mass with respect to $NO_x$ mass is calculated. We choose the $NO_x$ to $O_x$ relationship as a way of evaluating the model given its central role in multiple atmospheric chemical pathways in





both the stratosphere and the troposphere. This column model test is considered appropriate as no changes have been introduced to the global transport, convection or mixing processes.

Forward model sensitivities, $\Lambda$, are obtained using the finite difference (brute force) method, with a single-sided finite
difference equation

$$\Lambda = \frac{J(x_0+h) - J(x_0)}{h}, \tag{1}$$

where $J$ is the objective (cost) function, $x_0$ is the baseline state of the model, and $h$ is the perturbation size. We evaluate these sensitivities at a tropospheric altitude of 3.9 km (625 hPa – model layer 20) to ensure that the tropospheric adjoint function is maintained, and at a stratospheric altitude of 21 km (44 hPa – model layer 40) to ensure the functioning of the stratospheric
additions. The perturbation size, $h$, is chosen balancing the effects of non-linearity of the response and the numerical round-off effects. On the one hand, a large $h$ may result in a deviation off the point at which the finite difference sensitivity is evaluated and, in the case of a non-linear response, provide an inaccurate estimate of the sensitivity. On the other hand, a small $h$ may result in subtraction round-off errors.

Figure 3 presents the sensitivity comparisons for each point in the global domain for an $h$ value of 100 kg/grid-cell and 300
kg/grid-cell for the tropospheric and stratospheric layer respectively. We find that these $h$ values balance the numerical artefacts of non-linearity and round-off error effects when calculating the finite difference sensitivity $\Lambda$. In both stratospheric and tropospheric level objective functions the gradients agree with $R^2 > 0.998$, with points off the regression line representing highly non-linear regimes. The off-diagonal cluster of points consists of the Southernmost grid cell row. While this evaluation
is performed on an individual chemistry time-step with horizontal transport processes disabled, it allows the simultaneous evaluation of the sensitivities for a wide range of different background conditions, including varying $NO_x$ levels (right column in Figure 3).



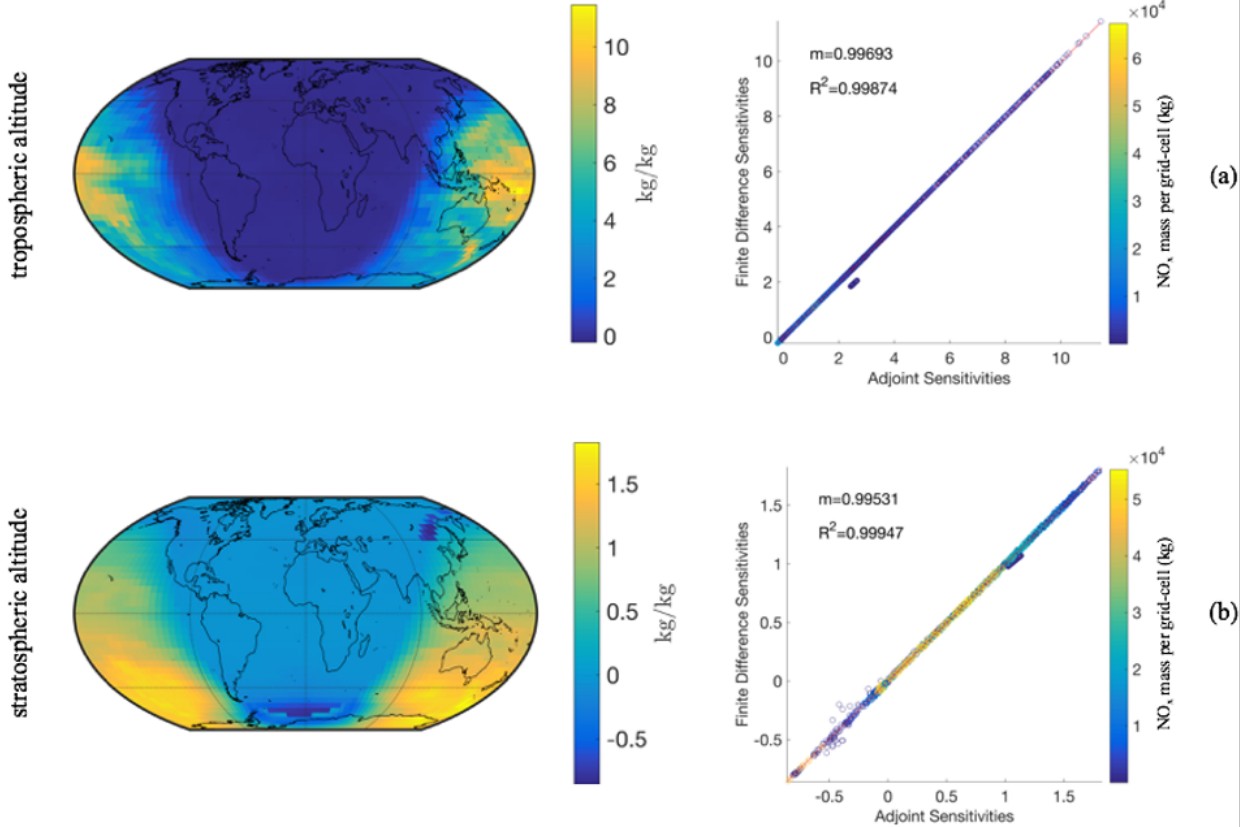

**Figure 3 Adjoint model evaluation after a single chemistry time-step (1 hour) with disabled transport and convection processes. In the left column the adjoint sensitivities depict changes in $O_x$ mass for $NO_x$ perturbations in the same grid-cell. In the right column are the adjoint gradients compared to finite difference gradients, with the corresponding linear regression slope, *m*, and coefficient of determination, $R^2$, for all *N* column models tested simultaneously ($N=46 \times 72$). Each point on the parity plot is coloured according to the $NO_x$ mass in the grid-cell. Top row (a) shows a tropospheric model layer (4 km) and bottom row (b) shows a stratospheric model layer (21 km).**

The effects of the choice of perturbation size, $h$, for the tropospheric and the stratospheric level sensitivity evaluations are presented in Figure 4. The clusters of off-diagonal points, which drive the $R^2$, move closer to the diagonal as $h$ decreases, indicating that these off-diagonal points represent non-linear regimes, in both the stratospheric and tropospheric comparison. At the same time however, we note more numerical noise for smaller $h$ values. For example, in the stratospheric case (panels c-e) the smaller the $h$, the higher the $R^2$ (driven by a larger number of non-linear points), however the more numerical noise points are evident (for sensitivities <0) compared to the cases with the larger $h$.

**Geoscientific Model Development Discussions**

a)   $h$ = 500 kg/grid-cell            b)   $h$ = 100 kg/grid-cell

c)   $h$ = 5000 kg/grid-cell        d)   $h$ = 500 kg/grid-cell        e)   $h$ = 300 kg/grid-cell

**Figure 4: Adjoint model evaluation after a single chemistry time-step (1 hour) with disabled transport and convection processes for different NO$_x$ perturbation sizes, $h$. Top row (panels a, b) displays the evaluation at a tropospheric model layer (4 km) and bottom row (panels c, d, e) at a stratospheric model layer (21 km). The adjoint gradients compared to finite difference gradients, with the corresponding linear regression slope, $m$, and coefficient of determination, $R^2$, for all $N$ column models tested simultaneously ($N$=46 × 72). Each point on the parity plot is coloured according to the NO$_x$ mass in the grid-cell.**

The base (forward component) model evaluation presented in 3.1, together with the component-level evaluation of the new/updated modules as well as the whole-model single timestep column evaluations presented here of the differentiated counterpart provide confidence in the correct implementation of the UCX adjoint development. While this is considered sufficient for the short-term sensitivity applications in stratospheric ozone described in the upcoming section, long term





sensitivity calculations would be necessary to capture the full effects of tropospheric-stratospheric exchanges. Additional changes necessary to enable long-term evaluations are described in (Fritz et al., 2022).

## 4 Model application

Using the updated tropospheric-stratospheric capabilities of the GEOS-Chem adjoint, we calculate short-term (two-week) ozone sensitivities to ozone depleting substances and precursors. An individual run of the GEOS-Chem UCX adjoint quantifies the relationship between model parameter perturbations and a scalar quantity of interest (objective function). Here we provide three examples that illustrate the information provided by the adjoint sensitivities, aiming to demonstrate the extended capabilities of the developed model to capture stratospheric ozone depletion, and its potential for providing an alternative way of examining the underlying chemical processes. For the following simulations, we use the global $4° × 5°$ global horizontal resolution (latitude × longitude) and 72 hybrid sigma-eta pressure levels extending from the surface to 0.01 hPa, driven by the GEOS5 assimilated meteorological data from the Global Modeling and Assimilation Office (GMAO) at the NASA Goddard Space Flight Center. We use the spun-up initial conditions referred to in Section 3.1for each simulation, ensuring that the concentrations of species (including reservoir species) are spatially and temporally appropriate. We run the GEOS-Chem adjoint for two-week intervals. This timescale is sufficient for capturing chemical relationships between ozone and short-term catalytic loss agents (e.g. active halogen and $NO_x$ species) at the corresponding altitudes and times of the year. We perform the simulations for odd oxygen as an objective function (numerator of sensitivity), for 1-15 of March and 1-15 of September of 2010, to capture the polar ozone depletion phenomena. We also use objective functions of stratospheric 'activated' and 'unreactive' chlorine to better describe the drivers behind the Antarctic ozone sensitivities calculated.

Figure 5 depicts the sensitivities of $O_x$ at a stratospheric vertical layer of the model (layer 40) ranging between 20.9 and 22.0 km (47.6 hPa and 40.2 hPa) with respect to perturbations in the $NO_x$ and Cl mass at the same layer. Given the receptor-oriented nature of the adjoint method, the maps indicate how a perturbation in the $NO_x$ and Cl mass anywhere in the domain would affect the aggregate $O_x$ at the same vertical model layer (i.e. there is no spatial information on the resulting ozone changes). These are provided for March and September. During the Antarctic spring in September, the ozone depletion potential is highlighted, with ~5 times greater magnitude sensitivities of $O_x$ to active chlorine (of which Cl is shown here), consistent with the observed high rates of heterogeneous chlorine activation during this period (Solomon, 1999). The sensitivities of $O_x$ with respect to $NO_x$ are also higher in absolute terms in September. Closer to the Antarctic the sign is negative, and surrounding the hole it is positive, reflecting the bounding of the hole over the Antarctic. We do not observe any sensitivity changes in the Arctic ozone in March. This may be due to an underestimate of Arctic ozone depletion by the forward model (see Figure 1c), or due to the higher variability of Arctic ozone depletion, and the fact that 2010 was a relatively warm year in the Arctic with little NH polar cap ozone loss being observed (NASA Ozone Watch, 2018; Weber et al., 2018).



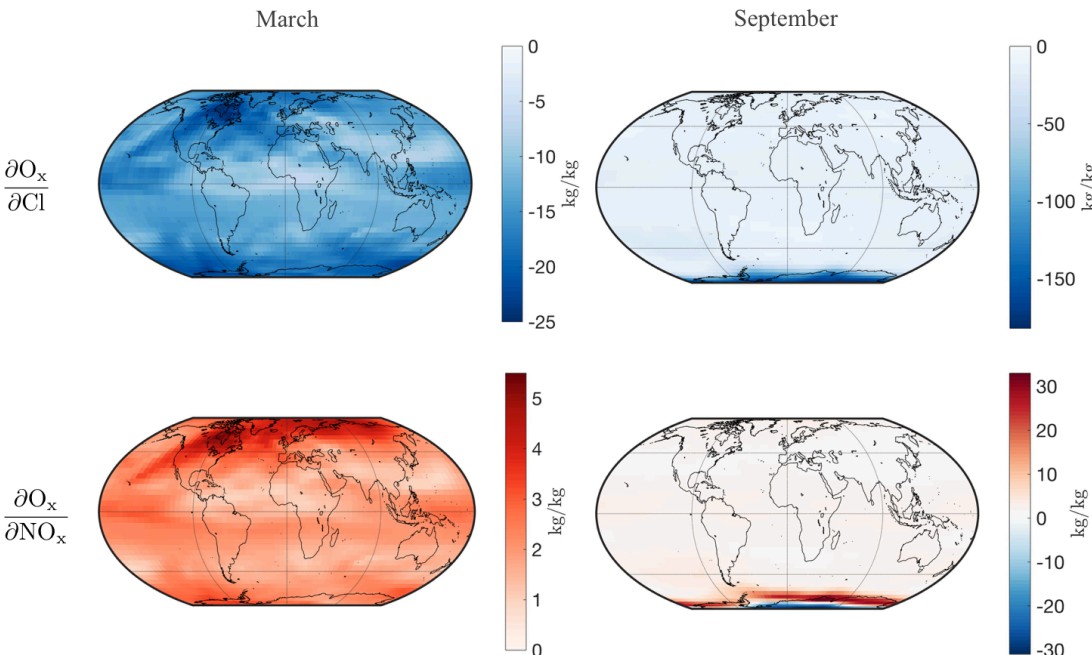

**Figure 5 Sensitivities of aggregate $O_x$ in a stratospheric vertical model layer (~21 km) with respect to perturbations in Cl (top row) and in $NO_x$ (bottom row) mass in the global domain of the same model layer, for a two-week simulation. Left column presents these for 1-15 of March and right column for 1-15 of September.**

The September $\partial O_x/\partial NO_x$ and $\partial O_x/\partial Cl$ sensitivities are shown in an Antarctic stereographic projection in Figure 6 (panel a and b respectively), together with the corresponding $O_x$ and $ClONO_2$ mixing ratios (panel c and d respectively). The sensitivities are largely bounded inside the ozone 'hole' (panel c). The rapid depletion of polar ozone which results in this ozone 'hole', occurs due to catalytic cycles in the sunlit atmosphere driven by activated forms of chlorine (Solomon, 1999).



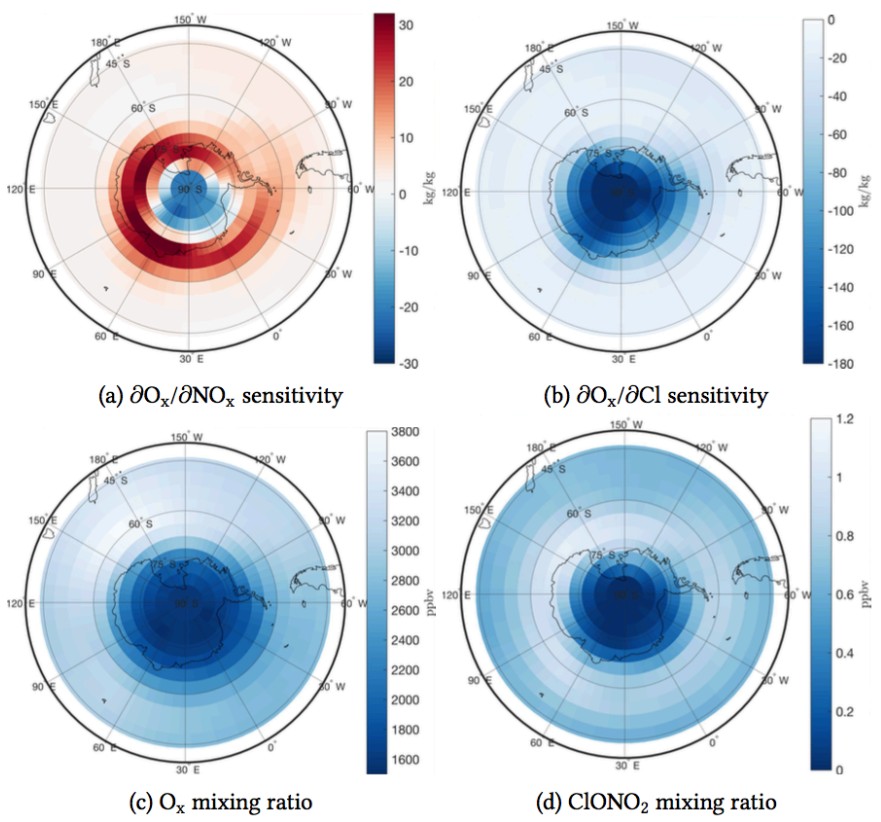

**Figure 6 Stereographic plots of $\partial O_x/\partial NO_x$ and $\partial O_x/\partial Cl$ sensitivities in kg/kg and $O_x$ and $ClONO_2$ mixing ratios in ppbv for a two-week September simulation at a stratospheric vertical model layer (~21 km).**

5  Chlorine, originating from compounds such as CFCs, exists at stratospheric altitudes in the form of inert reservoirs (e.g. $ClONO_2$ and HCl) referred to as 'unreactive' chlorine. Following heterogeneous processes, unreactive chlorine can convert into more active forms of chlorine (e.g. $Cl_2$, Cl, ClO, HOCl), referred to as 'activated' or 'reactive' chlorine. This partitioning of chlorine, as well as the activation and deactivation processes are central for understanding the polar ozone depletion mechanism (Solomon et al., 2015).

We find a high $\partial O_x/\partial Cl$ sensitivity inside the ozone 'hole' with a gradient towards the pole, illustrative of the odd chlorine catalytic cycle. In addition, the capability of $NO_x$ to neutralize the available ClO (Cl shown as a proxy to ClO, since these cycle rapidly) into unreactive $ClONO_2$ varies from the pole to the edge of the ozone hole. Its sign reversal forming a positive sensitivity 'collar' at the edge of the vortex links to the behaviour of the Antarctic $ClONO_2$ 'collar', which is visible in Figure

15  6d (Toon et al., 1989; Jaeglé et al., 1997; Chipperfield et al., 1994). Figure 7 shows the sensitivity of two chlorine deactivation and activation pathways, $\partial ClONO_2/\partial NO_x$ and $\partial Cl_2/\partial HCl$ respectively. The deactivation 'collar' in $\partial ClONO_2/\partial NO_x$ is likely the cause of the reversal in sign of the $\partial O_x/\partial NO_x$ sensitivity. The ozone loss, overlapping with the high chlorine activation



region in Figure 7b, is bounded over the Antarctic by this 'collar' as active chlorine is converted back into the $ClONO_2$ unreactive reservoir.

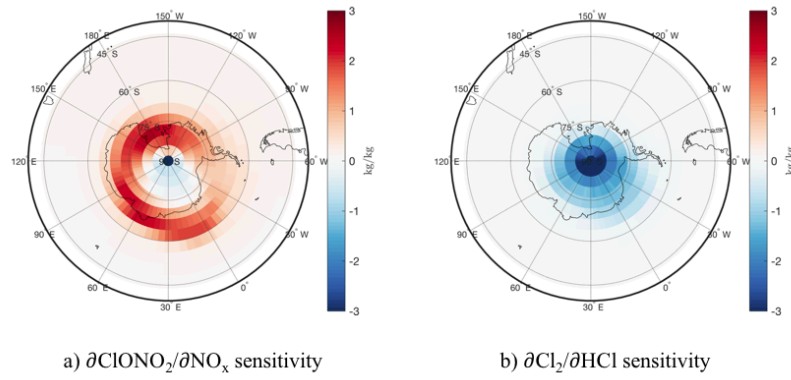

a) $\partial ClONO_2/\partial NO_x$ sensitivity          b) $\partial Cl_2/\partial HCl$ sensitivity

**Figure 7: Antarctic chlorine activation (a) and deactivation (b) adjoint sensitivities for a two-week September simulation at a**
5   **stratospheric vertical model layer (~21 km).**

Figure 8 shows the zonally averaged sensitivities of $O_x$ mass at stratospheric altitudes between 20 and 30 km with respect to active halogen species mass, BrO and ClO, for the first two weeks of March and September of 2010. Perturbations in ClO and BrO mass in all cases and at all altitudes lead to ozone depletion (negative sensitivity sign), reflecting the influence of catalytic
10   halogen cycles (Solomon, 1999). As previously mentioned, no clear Arctic ozone depletion potential is obtained in this case, potentially due to the climatology that year. In the Antarctic region in September the sensitivities highlight the altitude and latitude area where the ozone hole appears. We note that the sensitivity of $O_x$ with respect to BrO is ~15 times higher than the corresponding ClO one for both March and September on a per kg basis, in line with previous estimates of ~45 on a per atom basis (Daniel et al., 1999).



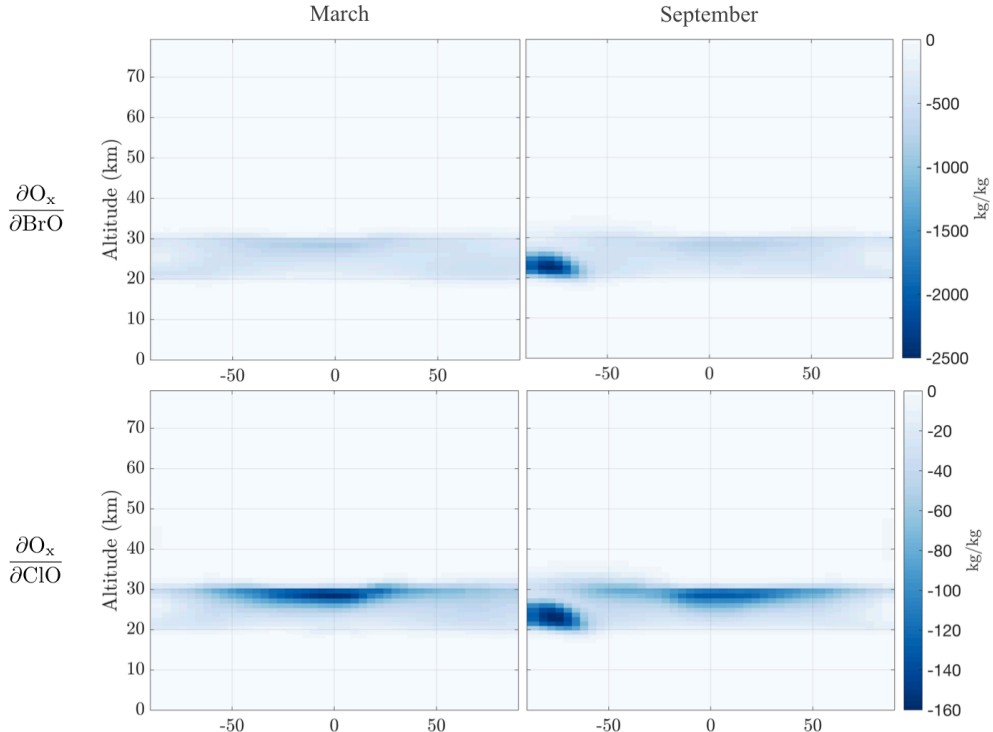

**Figure 8 Zonal sensitivities of aggregate $O_x$ at a stratospheric altitude band between 20 and 30 km with respect to perturbations in BrO (top row) and ClO (bottom row) mass in all domain altitudes, for a two-week simulation. Left column presents these for 1-15 of March and right column for 1-15 of September.**

The receptor-oriented adjoint sensitivities allow us to examine relationships between ozone and its precursors, by quantifying the effects of perturbations at different locations in the atmosphere. Given the interest in high-altitude $NO_x$ perturbations from the potential re-introduction of supersonic civil aircraft, Figure 9 shows zonally averaged sensitivities with objective functions of $O_x$ at 10-20 km and 20-30 km with respect to $NO_x$ mass perturbations anywhere in the domain during the two weeks in March 2010. Similar to the previous plots, these figures do not indicate where the ozone changes due to the $NO_x$ perturbation are occurring; instead they indicate how zonal $NO_x$ perturbations at different altitudes affect the aggregate $O_x$ at each respective altitude band. The sign of this sensitivity reverses between the two altitude bands, with $NO_x$ perturbations leading to increases in $O_x$ in the lower (10-20 km) region through well-known 'smog' chemistry but decreases in $O_x$ at higher (20-30 km) altitudes via $NO_x$-catalysed ozone destruction. The role of tropical convection is captured in the 10-20 km $O_x$ region, as emissions of $NO_x$ even near the surface can lead to increases in the 10-20 km $O_x$ mass. While some atmospheric dynamics effects (or the onset thereof) are captured in the 10-20 km sensitivities, no clear (i.e. external to the 20-30 km region) effects are present in the 20-30 km region in this 2-week simulation, reflecting the different transport timelines in the various regions of the atmosphere. Finally, the sign reversal of the sensitivity at different altitude bands implies the expected existence of an ozone neutral sensitivity regime, at which level the emissions of $NO_x$ perturbations would have a net zero effect on stratospheric



ozone. However, multiyear-long simulations would be required to capture the full tropospheric-stratospheric interactions, including the different transport time scales at different regions of the atmosphere (Fritz et al., 2022). The $\partial O_x/\partial NO_x$ sensitivity at additional altitude bands is described in Section S3 in the SI.

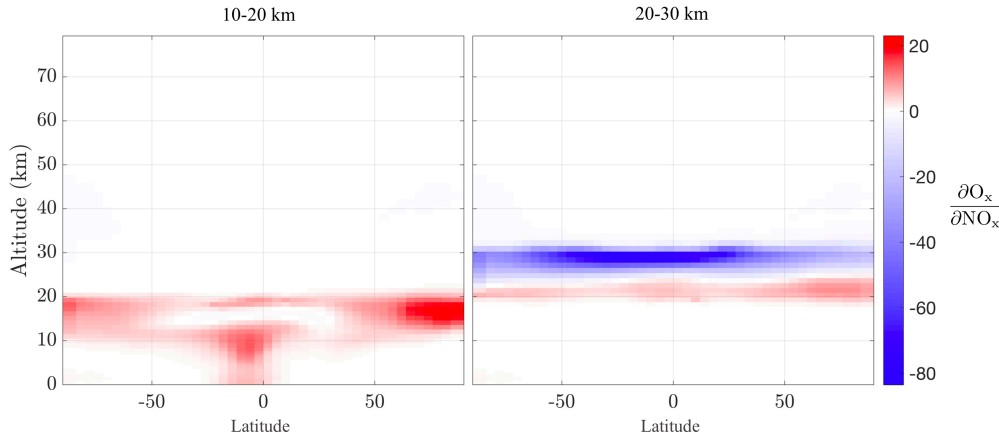

**Figure 9 Zonal sensitivities of aggregate $O_x$ at two stratospheric altitude bands (10-20 km and 20-30 km) with respect to perturbations in $NO_x$ mass in all domain altitudes in kg/kg, for a two-week simulation in March.**

## 5 Summary and conclusions

Sensitivity analyses are widely used in quantifying source-receptor (or source-effect) relationships, and further applied in gradient based assimilation and optimization. Stratospheric sensitivities can be used to enhance the understanding of underlying chemistry and physics from a new perspective and provide insight on how emissions or other atmospheric changes may lead to ozone depletion. Obtaining sensitivities to all parameters using traditional modelling approaches is computationally intractable. Adjoint, receptor-oriented sensitivities overcome this computational cost, under the assumption that the number of sources of interest is significantly greater than the number of receptors.

This work describes the development of the adjoint of the global GEOS-Chem unified tropospheric-stratospheric chemistry extension (UCX) CTM, which extends the tropospheric capabilities of the GEOS-Chem adjoint (prior to v36) to include stratospheric chemistry. The adjoint model is validated against finite difference tests of the forward component model. We apply the GEOS-Chem UCX adjoint model to calculate short-term stratospheric ozone receptor-oriented sensitivities to ozone (production and loss) precursors. We quantify the Antarctic ozone depleting potential of BrO, ClO, Cl, and $NO_x$, as well as the altitude dependence of the $NO_x$ to $O_x$ production/loss relationship.

In this paper we use stratospheric $O_x$ to demonstrate the capabilities of the model in providing a new perspective for examining the underlying chemical and physical processes in a receptor-oriented way. However, sensitivities of any tracer to model





parameters can be computed. As such, the adjoint of the GEOS-Chem UCX can be applied to assess the impacts of, including but not limited to, volcanic emissions, changes in water vapor, as well as stratospheric-tropospheric exchanges. Additionally, besides the largely chemistry driven phenomena captured in the two-week sensitivities presented in this work, longer runs would yield insight on the coupled transport and chemistry phenomena. Longer-term sensitivities would also capture the ozone

layer impacts of ground-level emissions perturbations. The adjoint of GEOS-Chem UCX also enables the assimilation of observations in an inverse modelling framework, and thus the potential for addressing a wide range of scientific questions.

*Acknowledgements.* We thank Susan Solomon for helpful discussions. This work was supported by NASA under cooperative agreement NNX14AT22A. ICD was also funded through the MIT Martin Family Fellowship for Sustainability.

*Author contribution.* Conceptualization: SB; Funding acquisition: SB.; Investigation: ID, DH, SE, RS, SB; Methodology: ID, DH; Project administration: RS, SE, SB; Software: ID, DH; Supervision: SB; Visualization: ID; Writing–original draft: ID; Writing–review and editing: all

*Code availability.* The source code for GEOS-Chem adjoint model v36 is openly available and instructions about accessing the code and the required inputs can be found at: https://wiki.seas.harvard.edu/geos-chem/index.php/GEOS-Chem_Adjoint. This model development and application is undertaken in the GEOS-Chem adjoint version v35f (available at: 10.5281/zenodo.4300535) and incorporated in the openly available GEOS-Chem adjoint v36. More details on the GEOS-Chem adjoint versions can be found here: http://wiki.seas.harvard.edu/geos-chem/index.php/GEOS-

Chem_Adjoint#Current_GEOS-Chem_adjoint_version_released.

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
