# Peer review of "Development of the adjoint of the unified tropospheric-stratospheric chemistry extension (UCX) in GEOS-Chem Adjoint v36"

_Geoscientific Model Development, 2023_

## Author Comment (AC1)

Dear Editor and referees,

Re: Submission of manuscript titled "Development of the adjoint of the GEOS-Chem unified tropospheric-stratospheric chemistry extension (UCX) in GEOS-Chem Adjoint v36" to GMD.

Thank you for arranging review of our manuscript. Below, you will find our response to the referees, answering each comment, clarifying our approach and detailing how their suggestions were incorporated in the revision (referee comments in *italics*, our response **in bold** font, page/line numbers referring to the updated manuscript).

Based on the referee suggestions, we have expanded the description of the model development and clarified the model evaluation aspects. We believe that these changes have improved the quality of our manuscript.

Thank you again for considering our submission to GMD, and we look forward to your response.

Sincerely,

Irene Dedoussi

**Referee #1**

*Adjoint models are highly effective tools for identifying sensitivities in physical processes, making their development invaluable to the scientific community. The authors have commendably demonstrated how the proposed adjoint model can be utilized to explore sensitivity information, suggesting its potential applicability across a broad spectrum of use cases. However, my primary concern with the study lies in the validation of the developed adjoint model, as detailed below. Addressing the following two points would make me more inclined to recommend this work for publication:*

**We would like to thank the referee for the appreciation of the adjoint modelling approach and the feedback.**

*1. The validation of adjoint sensitivities against forward modeling results in this study is conducted qualitatively. To establish the adjoint's accuracy quantitatively, it is essential to validate the model using the relationship $\langle \Delta x\_1, M\Delta x\_2 \rangle = \langle M^T\Delta x\_1, \Delta x\_2 \rangle$, where x\_1 and x\_2 represent any state vectors, M symbolizes the tangent linear of the forward model, and M^T denotes the adjoint. The left-hand side (LHS) and right-hand side (RHS) of this equation should match to machine accuracy (e.g., 13 or more digits in double precision) to ensure reliability. Without this level of agreement, the results, while appearing reasonable in certain instances, cannot be universally trusted.*

**We would first like to clarify that we do not use (nor have implemented) the tangent linear model of the forward model, as we developed the adjoint directly. Instead, to validate the model development undertaken in this work, we compare the newly developed adjoint UCX model against finite difference sensitivities from the forward GEOS-Chem model. In this way we are evaluating, quantitatively, the adjoint's performance directly against the non-linear forward model. In the manuscript, this comparison is presented for the entirety of the adjoint model covering all processes excluding transport processes which have not been changed in this development (to be able to perform this evaluation for all model columns in one simulation). Additionally, we note that during the model development this comparison was also performed in a component-wise manner for each individual adjoint (sub)routine and (sub)processes developed (listed in Table S2).**

**We have added a sentence clarifying this in the manuscript (p. 9, lines 2-4).**

*2. Demonstrating the validity of the tangent linear approximation within the context under study is crucial before employing the adjoint model to derive sensitivities. This step is fundamental to ensuring that the adjoint sensitivities are meaningful. A practical approach to validate this would be to compare the outcomes of the tangent linear model with those from nonlinear forward model differences, thereby reinforcing the model's credibility.*

**Similarly, to the response to the previous comment, here are comparing the adjoint against nonlinear forward model differences, thereby effectively also addressing concerns regarding the potential nonlinearity of the model. Additionally, since we have examined the impact of different finite difference perturbation sizes (Figure 4), we do have some understanding of the degree of nonlinearity of the forward model. As shown in Figure 4, the model is strongly nonlinear in a cluster of grid cells, but overall demonstrates decent linearity for the perturbation steps tested. However, we note that the degree of linearity also depends on the altitude (Fritz et al. 2022).**

*Addressing these concerns will significantly strengthen the manuscript, making it a robust contribution to the field.*

**Referee #2**

*This paper is well in the scope of GMD for its model development of UCX in GEOS-Chem Adjoint.*

*My main concern for manuscript is that for a model development paper, the methods part - including model description and especially model development - is lacking. As a GEOS-Chem modeler myself, I cannot reproduce the authors' work based on the two paragraphs currently present in the paper. The development description is too general and vague to be replicable. It could benefit from more details, especially for a model development publication.*

**We would like to thank the reviewer for the constructive feedback. We have expanded the Model development section (Section 2.2) with additional details on how the adjoint model is developed. For the capabilities of the forward (based) model we now direct readers to the extensive description of that development in Eastham et al. (2014). For the adjoint model development we have expanded the section in order to provide more details on both the development choices as well as the software engineering aspects of it. Specifically, we have expanded on the motivation for the discrete adjoint choice, the checkpointing (forward model values) that is needed, and the implementation of the adjoint model development.**

**The relevant text is in Section 2.1 (p. 5, lines 5-6) and Section 2.2 (p. 5, 17-20, 26-32; p. 6, lines 1-6).**

*Two other minor suggestions:*

*The title is repetitive, and I advise a more succinct "Development of the unified tropospheric- stratospheric chemistry extension (UCX) in GEOS-Chem Adjoint v36" would get the point across sufficiently. But this may be a stylistic choice and merely a suggestion.*

**Thank you for this suggestion. We have reworded the title to: "Development of the adjoint of the unified tropospheric-stratospheric chemistry extension (UCX) in GEOS-Chem Adjoint v36".**

*Most figures could also benefit from better pixels for publication, for example, Figs 6 and 7's fonts are small, and once enlarged, they're quite blurry.*

**We have updated the size and resolution of Figure 6 and Figure 7.**

*The rest of the paper is scientifically sound and well presented. Once more details for the methods are added, I would support its publication in GMD.*

References:

Eastham, S. D., Weisenstein, D. K., and Barrett, S. R. H.: Development and evaluation of the unified tropospheric–stratospheric chemistry extension (UCX) for the global chemistry-transport model GEOS-Chem, Atmospheric Environment, 89, 52–63, https://doi.org/10.1016/j.atmosenv.2014.02.001, 2014.

Fritz, T. M., Dedoussi, I. C., Eastham, S. D., Speth, R. L., Henze, D. K., and Barrett, S. R. H.: Identifying the ozone-neutral aircraft cruise altitude, Atmospheric Environment, 276, 119057, https://doi.org/10.1016/j.atmosenv.2022.119057, 2022.